# Theoretical Analysis and Experimental Research on the Adjustment for Pre-Stress Deviation of the Cable-Bar Tensile Structures

**Lianmeng Chen** [1,*]**, Yijie Liu** [1]**, Yihong Zeng** [1]**, He Zhang** [1] **and Yiyi Zhou** [2]

[1] College of Civil Engineering and Architecture, Wenzhou University, Wenzhou 325035, China; 194611571239@stu.wzu.edu.cn (Y.L.); 20461542066@stu.wzu.edu.cn (Y.Z.); 184611572178@stu.wzu.edu.cn (H.Z.)

[2] College of Civil Engineering and Architecture, Changzhou Institute of Technology, Changzhou 213002, China; zhouyy@czu.cn

[*] Correspondence: 00151034@wzu.edu.cn; Tel.: +86-139-5779-0090

**Abstract:** Construction errors are unavoidable in actual cable-bar tensile structures. Construction error analysis, evaluation, and especially adjustment theories were still in their infancy. For the improvement of the situation, based on the equilibrium equation, physical equation, and geometric equation for pin-joint structures, the member length deviation was adopted as the variable, and the relationship between the pre-stress deviation and member length deviation was determined. On this basis, an adjustment method was devised for the pre-stress deviations under three different conditions, and an evaluation of the effectiveness for pre-stress deviation adjustment was proposed. Finally, a 5-m diameter cable-bar tensile structure model was designed and constructed for simulation. The research results demonstrated that the adjusted pre-stress deviations of measuring points can be effectively corrected, and the theoretical results generally coincided with the experimental results. The adjustment effects of pre-stress deviation varied with the number of adjustment cables, and the adjustment effectiveness gradually decreased with the reduction of the number of adjustment cables. Different adjustment schemes produced different structural deformations, and it was necessary to prioritize the adjustment scheme that resulted in lower peak values of internal forces and shape changes during the adjustment process. The research results indicated that the correctness and validity of the proposed error analysis and adjustment method of pre-stress deviation, and its practical application in the guidance of construction errors analysis, pre-stress deviation adjustments, and evaluation of adjustment results of actual pretension structures.

**Keywords:** cable-bar tensile structures; construction error; pre-stress deviation adjustment; evaluation of adjustment results; model research

## 1. Introduction

The cable-bar tensile structure is a sort of flexible spatial structure composed of tension cables and compression bars with a tension-forming and self-balancing pre-stress system. Due to the full utilization of high-strength cables and the capability of adjusting pre-stress distribution to optimize stiffness distribution of the structure, this type of structure is characterized by large span, light weight, economical cost, and other advantages, and has been widely used in practical engineering [1–4], such as the typical cable-bar tensile structures for the 1988 Seoul Olympics and 1996 Atlanta Olympics, as shown in Figures 1 and 2. Before the construction of this sort of structure, the pre-stress is zero. With the progression of construction, the pre-stress will increase gradually and arrive the final pre-stress distribution upon the end of the construction. Therefore, the precise pre-stress distribution of the cable-bar tensile structure is the prerequisite of excellent mechanical performance. Nevertheless, because of the complex construction conditions and other negative impacts,

construction errors are inevitable, including member length error, installation error, node position error, etc. Consequently, the real parameters will be deviated from the theoretical ones, for instance, the pre-stress distribution and node position [5,6]. Previous studies have shown that the mechanical performance of the structure is affected sensitively by the pre-stress deviation [7–9], so it is valuable to assess the effects of different construction errors, especially when exploring the adjustment methods for construction errors and pre-stress deviations so as to lessen the negative effects.

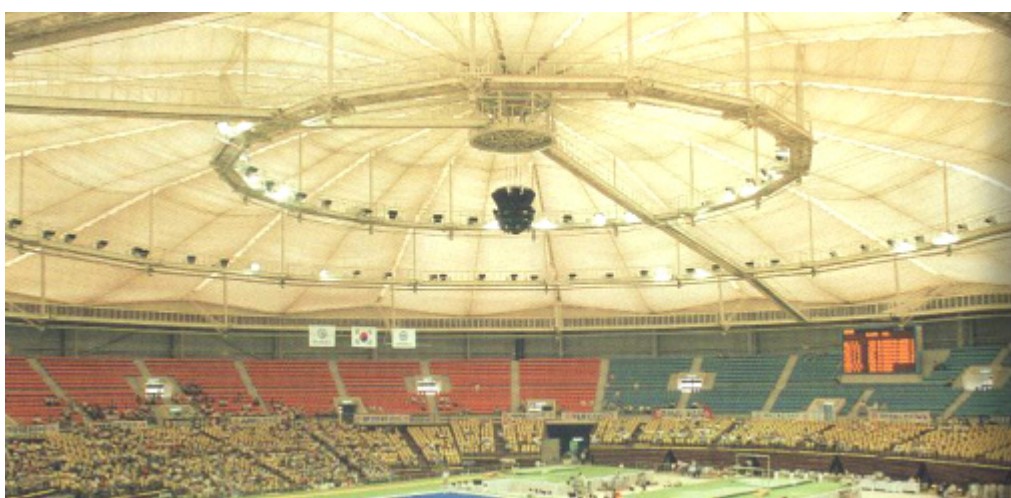

**Figure 1.** A cable-bar tensile structure for Seoul Olympics.

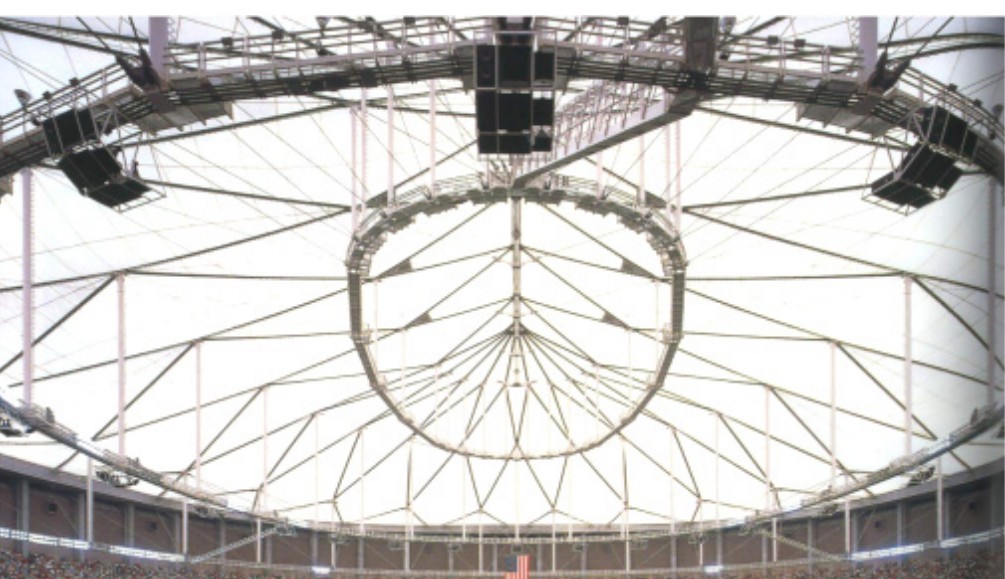

**Figure 2.** A cable-bar tensile structure for Atlanta Olympics.

In the last two decades, worldwide research efforts were underway to develop adjustment methods for construction errors and pre-stress deviations of large-scale structures, such as spatial large-span structures and large-span bridge structures. Hao and Ding [10] put forward a correction method of pre-stress deviation in cable-stayed bridge construction by applying the least square method and considering the nonlinear characteristics of structure. Weng and Xiang [11] adopted the least square method to propound the correction method for pre-stress deviation of the cable-stayed Wenhui Bridge in Hangzhou City. Based on the node displacement correction method, Zhang [12] adopted the cyclic iterative solution method to obtain the adjustment values of the suspend-dome structure with pre-stress deviation. On the basis of the pretension adjustment method, Zhuo [13] adopted

the cyclic iterative solution method to obtain the adjustment values of the suspend-dome structure with pre-stress deviation. Based on the measured data of the experimental model, Guo [14] adopted the two-dimensional search method to adjust the boundary constraint conditions of the calculation model of the suspend-dome structure. Zhang [15] based his research on the random theory to analyze error sensitivity and rank the elements in the sequence of error sensitivities. As the element was sensitive to its pre-stress deviation, its length was adjusted preferentially to realize the adjustment of the element's internal force. Yet, this method worked merely as a qualitative guidance to adjust pre-stress deviations and required repeated adjustments. Yu et al. [16] did a case analysis of Yueqing Sports Center Stadium with the crescent-shaped cable-bar tensile structure and proposed the continuous adjustment of pre-stress deviation by adopting the method of nonlinear finite element. Taking a more than 100-m-span cable dome in China as a case study, Chen [17] adopted the outmost ridge cables and outmost diagonal cables as the adjustment cables to eliminate the construction errors of the ring beam and optimized the field placement to reduce the random cable errors. It can be seen that studies in relation to construction error evaluation and adjustment of both large-span structures and spatial pre-stress structures are underway and have made certain research achievements. Yet, there is still a lack of research regarding the construction errors analysis, pre-stress deviation adjustment, and evaluation of adjustment results for the flexible cable-bar tensile structure.

Considering the fact that there was a lack of an efficient method to analyze and evaluate the construction errors of the cable-bar tensile structure, and especially no proper adjustment method was available to correct these errors, based on the previous basic work by the authors, the member length deviation was adopted as the variable to propose a method to correct the pre-stress deviations. Firstly, based on the equilibrium equation, physical equation, and geometric equation for pin-joint structures, a relationship between the pre-stress deviation and member length deviation was built. On this basis, a correction method was devised for the pre-stress deviation under various schemes, and an evaluation of the efficiency for pre-stress deviation correction was proposed by using measured data. Finally, a 5-m diameter cable-bar tensile structure model was designed and constructed for simulation, in which the pre-stress deviation of the measuring point was adjusted by adjusting the length of the outer diagonal cable in stages. Then, the theoretical values and the measured values of the measuring points during the adjustment process were compared and evaluated. On this basis, the adjustment effects of different adjustment schemes on the pre-stress deviation and shape deviation were further analyzed, compared, and evaluated. Thus, this paper proposes a theoretical foundation for the construction errors analysis, pre-stress deviation adjustment, and evaluation of adjustment results of actual pretension structure, and offers valuable insights into both theoretical research and actual engineering application.

## 2. Fundamental Theory of the Relation between Pre-Stress Deviation and Member Length Deviation

In order to analyze the effects of construction errors, member length error, which is a significant construction error and will affect the mechanical performance sensitively, was chosen as the representative construction error, and the relation between the pre-stress deviation and member length error needs to be clarified first. In a cable-bar tensile structure, the following three equations can be established [18,19].

$$A_{3n \times b} t_{b \times 1} = P_{3n \times 1} \tag{1}$$

$$t_{b \times 1} = M_{b \times b} \left( e_{b \times 1} - (e_0)_{b \times 1} \right) \tag{2}$$

$$B_{b \times 3n} d_{3n \times 1} = e_{b \times 1} \tag{3}$$

In the Equations (1)–(3), $A_{3n \times b}$ and $B_{b \times 3n}$ are the equilibrium matrix and the coordinates matrix, respectively; $B = A^{\mathrm{T}}$, $t_{b \times 1}$ is the internal force matrix; $P_{3n \times 1}$ and $d_{3n \times 1}$ are the nodal load matrix and node displacement matrix, respectively; $M_{b \times b}$ is the member

stiffness matrix; $M_{kk} = E_k A_k / l_k$, $E_k$, $A_k$ and $l_k$ are the elastic modulus, section size, and length for member $k$, respectively; $e_{b \times 1}$ and $(e_0)_{b \times 1}$ are the member length change matrix and the initial member length change matrix, respectively; and $b$ and $n$ are the total number of member and free node. The combinations of Equations (1)–(3) are expressed as follows:

$$t = t_P + t_e \tag{4}$$

$$d = d_p + d_e \tag{5}$$

In the Equations (4) and (5), $t_P = MA^T(AMA^T)^{-1}P$, $t_e = M(A^T(AMA^T)^{-1}AM - I)e_0$, $d_p = (AMA^T)^{-1}P$, and $d_e = (AMA^T)^{-1}AMe_0$, $I$ is the unit matrix; $t_P$ and $t_e$ are the member internal force resulting from nodal load $P$ and the initial member length defect $e_0$, respectively; and $d_P$ and $d_e$ are the node displacement resulting from nodal load $P$ and the initial member length defect $e_0$, respectively. Thus, when the nodal load $P$ remains unchanged or is zero, the member pre-stress deviations and node displacement deviations are led only by the initial member length variation. Suppose the member length changes by $\delta e_0$, the member pre-stress deviations and node displacement deviations are

$$\delta t = S_t \delta e_0 \tag{6}$$

$$\delta d = S_d \delta e_0 \tag{7}$$

In the Equations (6) and (7), $S_t = M(A^T(AMA^T)^{-1}AM - I)$ stands for the force sensitivity matrix and $S_d = (AMA^T)^{-1}AM$ stands for the shape sensitivity matrix. When the nodal load $P$ is not taken into account, the pre-stress is $t_0 = S_t e_0$.

According to the function of the members during the construction process, they can be sorted into two different kinds. One kind is active member that helps tension in the structure. The active member's length is corrected in the process of construction, and its internal force is read instantly by the jack. The other kind is passive member that has an already-known slack length, and its internal force will be generated gradually with the tension of the active member. To take into account both types of the above members, the initial pre-stress $t_0$ is then expressed as

$$t_0 = \left\{ \begin{array}{c} t_0^a \\ \hline t_0^p \end{array} \right\} = \left[ S_t^a \middle| S_t^p \right] \left\{ \begin{array}{c} e_0^a \\ \hline e_0^P \end{array} \right\} \tag{8}$$

where $t_0^a$ and $t_0^p$ represent the initial pre-stress led by initial member length defects in the active members and passive members; $S_t^a$ and $S_t^p$ represent the force sensitivity matrices for the active and passive members; $e_0^a$ and $e_0^p$ represent the initial length defects of the active and passive members, respectively. When the nodal load $P$ remains unchanged or is not taken into account, Equation (6) turns into

$$\delta t = S_t \delta e_0 = \{ S_t^a | S_t^p \} \left\{ \begin{array}{c} \delta e_0^a \\ \hline \delta e_0^p \end{array} \right\} = S_t^a \delta e_0^a + S_t^p \delta e_0^p \tag{9}$$

where $\delta e_0^a$ and $\delta e_0^p$ are the length deviations of the active and passive members.

## 3. Adjustment Method for the Pre-Stress Deviation

### 3.1. Fundamental Theory for the Adjustment of Pre-Stress Deviation

The adjustment of the pre-stress deviation is generally conducted by correcting the adjustment members. During the correction procedure, the adjustment members are monitored to guarantee the accurate correction value, and the optimized result of the correction is to make the pre-stress free from deviation. Suppose the number of adjustment

members is $k$. In order to achieve $\delta t_q = 0$, adjustment members must generate a length change, $\delta e_0^k$, and meet the following condition:

$$ -\delta t_q = S_{tq}^k \delta e_0^k \tag{10} $$

where $S_{tq}^k$ is the force sensitivity matrix for the adjustment members. If the values of the length change, $\delta e_0^k$, of the adjustment members can be obtained on the basis of the monitored values of the pre-stress deviation, $\delta t_q$, then the obtained $\delta e_0^k$ can be applied to correct the pre-stress deviation. To obtain $\delta e_0^k$, $\delta t_q$ in Equation (10) is added to the force sensitivity matrix, $S_{tq}^k$, to compose an augmented matrix, $\left[ S_{tq}^k | \delta t_q \right]$. Based on the matrix theory, the value of adjustment, $\delta e_0^k$, can be calculated by analysing the relationship between the rank $r_k$ of the matrix $S_{tq}^k$ and the rank $r_{k\prime}$ of the augmented matrix $\left[ S_{tq}^k | \delta t_q \right]$.

1.  If $r_{k\prime} = r_k < k$, then multiple solutions of $\delta e_0^k$ are present for Equation (10), and not all the adjustment members are needed to correct the pre-stress deviation. If $r_{k\prime} = r_k = q < k$, then only $q$ adjustment members are needed.
2.  If $r_{k\prime} = r_k = k$, then it means only one solution for $\delta e_0^k$ is available for Equation (10), and the unique solution can be obtained by

$$ \delta e_0^k = -[S_{tq}^k]^+ \delta t_q \tag{11} $$

    Equation (11) indicates that $k$ adjustment members are adequate to obtain a precise adjustment values of the pre-stress deviation. Moreover, the number of monitored cables, $q$, does not exceed the number of adjustment members, $k$. The correction values for the pre-stress deviation, $\delta \hat{t}^k$, can be obtained by substituting the values of the length change, $\delta e_0^k$, into Equation (10).
3.  If $r_{k\prime} = k + 1$, then it means no accurate solution for $\delta e_0^k$ is available for Equation (10); namely, the pre-stress deviation cannot be corrected only by correcting the $k$ adjustment members. Extra adjustment members are needed to precisely correct the pre-stress deviation, and the least square estimation is used to obtain approximate $\delta e_0^k$.

$$ (\delta e_0^k)^{\#} = [(S_{tq}^k)^{\mathrm{T}} (S_{tq}^k)]^{-1} (S_{tq}^k)^{\mathrm{T}} \delta t_q \tag{12} $$

    Then, the correction values for the pre-stress deviation, $\delta \hat{t}^k$, can also be calculated by substituting this estimate $(\delta e_0^k)^{\#}$ into Equation (10).

### 3.2. Evaluation of the Efficiency for Pre-Stress Deviation Adjustment

Based on the above method, the fitted values of $\delta \hat{t}^k$ can be obtained by substituting the adjustment of the cable length, $\delta e_0^k$, into Equation (10). If the monitored values are not equal to the fitted values, namely, $\delta \hat{t}^k \neq \delta t_q$, then the adjustment efficiency in adjusting the pre-stress deviation must be further assessed.

$$ (\delta e_0^k)^{\#} = [(S_{tq}^k)^{\mathrm{T}} (S_{tq}^k)]^{-1} (S_{tq}^k)^{\mathrm{T}} \delta t_q \tag{13} $$

where $\delta$ denotes the differences between the fitted and monitored values. Furthermore,

$$ (\delta e_0^k)^{\#} = [(S_{tq}^k)^{\mathrm{T}} (S_{tq}^k)]^{-1} (S_{tq}^k)^{\mathrm{T}} \delta t_q \tag{14} $$

where $\eta$ is the sum of squares of the deviations between the fitted and monitored forces. The smaller the value of $\eta$ is, the more efficient the correction is, and vice versa.

After adjusting the pre-stress deviation of the measuring point, the node displacement generated in the adjustment process should also be analyzed simultaneously; that is, the

shape deviation resulting from the pre-stress deviation adjustment. The cable length adjustment, $\delta e_0^k$, is obtained through Equation (11) or (12), and the displacement vector of each node in the adjustment process can be obtained, which is defined as:

$$\phi = (\delta \boldsymbol{d})^{\mathrm{T}} \cdot \delta \boldsymbol{d} \qquad (15)$$

where $\phi$ is the sum of squares of all the nodal displacements. It is evident that the smaller the shape change generated by the correction of member length, the more efficient the correction is, and vice versa.

## 4. Experimental Research

### 4.1. Model Design

In order to validate the proposed adjustment method of pre-stress deviation, a 5-m diameter cable-bar tensile structure model was designed and constructed, as shown in Figure 3. It was made up of 12 pieces of symmetrical cable-bar units, which contained cables and bars. The cables were categorized into three groups: hoop cables, ridge cables and diagonal cables. Hoop cables consisted of hoop cable 1 (denoted as HC1) and hoop cable 2 (denoted as HC2); ridge cables consisted of ridge cable 1 (denoted as RC1) and ridge cable 2 (denoted as RC2); and diagonal cables consisted of diagonal cable 1 (denoted as DC1) and diagonal cable 2 (denoted as DC2). Meanwhile, the compression bars consisted of Bar 1 and Bar 2. In this study, the 12 DC1 were selected as the active members, while the remaining were the passive members.

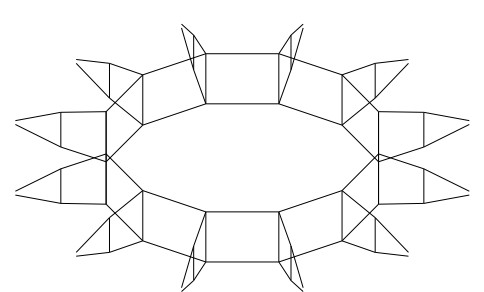

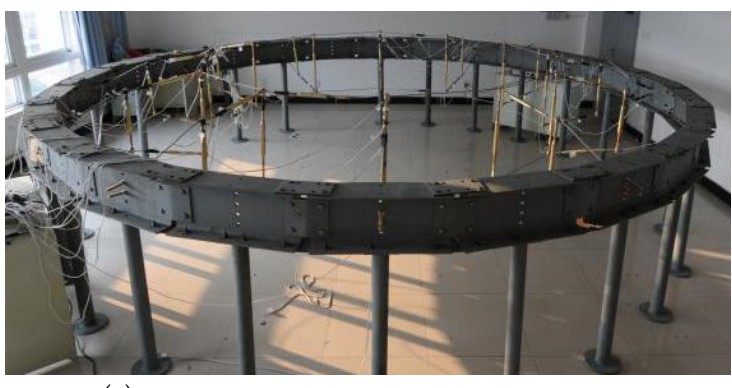

(a)

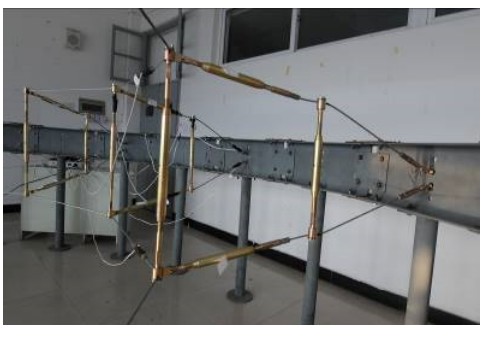

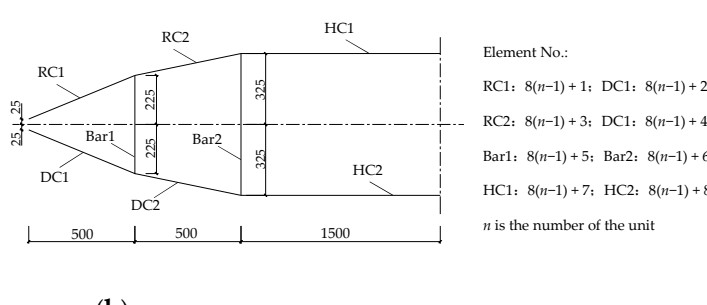

(b)

**Figure 3.** A cable-bar tensile structure model. (**a**) The whole model; (**b**) A symmetrical cable-bar unit.

During the experiment, all the 12 DC1 were also used to lay out the measuring points. The resistance foil strain gauge, BX120-5AA, was applied to monitor the force of the member, and a static data logger was utilized to obtain the strain. Prior to the experiment,

the relation between the force and the strain of the 12 DC1 must be defined, and the least square method was applied as follows:

$$N = k\varepsilon \tag{16}$$

where $N$ and $\varepsilon$ represented the force and the strain, and $k$ was the coefficient to be determined by Equation (17) and shown in Table 1, where the number n in the n-DC1 represented the unit number. Table 1 indicated that the relationship between force and strain of each monitoring point was nearly linear.

$$k = \frac{\sum_i N_i \varepsilon_i}{\sum_i \varepsilon_i^2} \tag{17}$$

**Table 1.** The coefficients k of 12 DC1.

| Loading Order | Weight (kg) | 1-DC1 | 2-DC1 | 3-DC1 | 4-DC1 | 5-DC1 | 6-DC1 | 7-DC1 | 8-DC1 | 9-DC1 | 10-DC1 | 11-DC1 | 12-DC1 |
|---|---|---|---|---|---|---|---|---|---|---|---|---|---|
| 0 | 0 | 0 | 0 | 0 | 0 | 0 | 0 | 0 | 0 | 0 | 0 | 0 | 0 |
| 1 | 20 | 3 | 4 | 4 | 3 | 4 | 3 | 4 | 3 | 4 | 3 | 3 | 3 |
| 2 | 40 | 6 | 6 | 7 | 6 | 7 | 6 | 7 | 6 | 6 | 6 | 7 | 7 |
| 3 | 60 | 10 | 9 | 10 | 10 | 10 | 10 | 10 | 9 | 10 | 10 | 11 | 10 |
| 4 | 80 | 14 | 12 | 14 | 14 | 13 | 13 | 14 | 13 | 13 | 14 | 14 | 14 |
| 5 | 100 | 17 | 15 | 17 | 16 | 17 | 16 | 17 | 17 | 17 | 16 | 17 | 17 |
| 6 | 120 | 20 | 18 | 20 | 20 | 20 | 19 | 21 | 20 | 20 | 20 | 21 | 21 |
| k | | 58.23 | 65.0 | 57.68 | 59.17 | 58.44 | 61.26 | 56.59 | 59.56 | 58.80 | 59.17 | 56.23 | 56.77 |

### 4.2. Experimental Process

After the tensioning and forming of the model, the pre-stresses in the 12 DC1 were measured through the measuring points. Then, it was found that the measured pre-stresses of the 12 measuring points differed from the theoretical values. In order to adjust the pre-stress deviation exactly, according to the above theory, at least 12 adjustment cables were needed, and the 12 DC1 were selected as the adjustment cables in this study; that is, the pre-stress deviations of 12 DC1 were corrected by adjusting the lengths of the 12 DC1. According to Equation (12), the variation value of each adjustment cable length, $\delta e_0^k$, can be obtained, where the negative values represented the decrease of element length and the positive values represent the increase of element length, as shown in Table 2.

**Table 2.** The pre-stress deviations of measuring points and adjustment values of cable length.

| Element No. | 2 | 10 | 18 | 26 | 34 | 42 | 50 | 58 | 66 | 74 | 82 | 90 |
|---|---|---|---|---|---|---|---|---|---|---|---|---|
| Theoretical value (kN) | 1.75 | 1.75 | 1.75 | 1.75 | 1.75 | 1.75 | 1.75 | 1.75 | 1.75 | 1.75 | 1.75 | 1.75 |
| Measured value (kN) | 1.63 | 1.99 | 1.90 | 1.57 | 1.86 | 1.94 | 1.52 | 1.58 | 1.90 | 1.53 | 2.06 | 1.95 |
| Deviation (%) | −6.86 | 14.00 | 8.63 | −10.17 | 6.29 | 10.80 | −13.37 | −9.43 | 8.80 | −12.46 | 17.83 | 11.31 |
| $\delta e_0^k$ (m) | −0.0030 | 0.0188 | 0.0103 | −0.0140 | 0.0055 | 0.0079 | −0.0281 | −0.0263 | −0.0023 | −0.0142 | 0.0257 | 0.0202 |

#### 4.2.1. Stages for the Pre-Stress Deviation Adjustment

1.  In this experiment, the length of DC1 was corrected by tightening or loosening the screw, and one end of the screw was connected to cable and the other end was connected to the supporting beam by a screw cap, as shown in Figure 4. In order to accurately obtain the adjustment length of the adjustment cable, it was necessary to begin with the measurement of the variation value of the cable length. $\Delta d = 2$ mm when the screw cap of each cable rotated a circle. The rotation angle was calculated according to the length adjustment value of each adjustment cable.
2.  According to the adjustment value of each DC1 in the sequence of number 2-10-18-26-34-42-50-58-66-74-82-90, the length of each DC1 was corrected accordingly, and the

forces of the 12 DC1 were recorded and compared with the theoretical values after the completion of adjustment of each stage. For convenience, this paper only recorded the values of internal forces of each measuring point at the completion of the four stages; namely, the lengths of cables 2, 10, and 18 were adjusted in the first stage, the lengths of cables 26, 34, and 42 were further adjusted in the second stage based on the first stage, the lengths of cables 50, 58, and 66 were further adjusted in the third stage based on the second stage, and the lengths of cables 74, 82, and 90 were adjusted in the final stage based on the third stage.

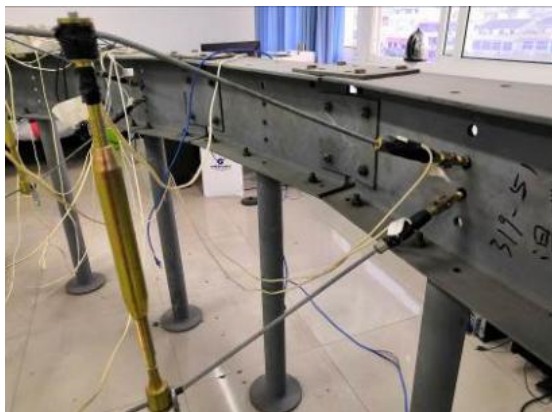 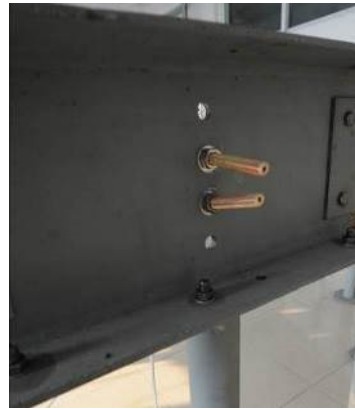

**Figure 4.** Screws connecting to cables and beam.

4.2.2. Result Analysis of Pre-Stress Deviation Adjustment

After the completion of adjustment of each stage, the theoretical values and measured values of internal forces of measuring points and their deviations were shown in Table 3.

1. The adjusted pre-stress deviation of each measuring point can be effectively corrected, suggesting the feasibility of error adjustment theory and its practical application to guide the adjustment of pre-stress deviation within the actual pretention structure construction.
2. In the adjustment process, although the deviations of some measuring points increased at certain stages, the pre-stress deviations of all measuring points gradually approached the ideal values until they were finally achieved.
3. Pre-stress deviations of measuring points for the 12 DC1 were not only caused by the length deviations of the 12 DC1, but also derived from the length errors of other elements, node errors, as well as adjustment errors which occurred during the process of the length adjustments of the 12 DC1. Thus, it was difficult to absolutely eliminate the pretention deviations, and there was still the presence of a few pre-stress deviations in the experiment.

**Table 3.** Adjustment to pre-stress deviation of the measuring point in four stages (kN).

| Adjustment Order | Measuring Point | 2 | 10 | 18 | 26 | 34 | 42 | 50 | 58 | 66 | 74 | 82 | 90 |
|---|---|---|---|---|---|---|---|---|---|---|---|---|---|
| The first-stage adjustment (2 + 10 + 18) | Theoretical value | −1.01 | −0.72 | −0.81 | −1.13 | −0.84 | −0.76 | −1.18 | −1.12 | −0.86 | −0.92 | −0.92 | −0.89 |
| | Measured value | −1.05 | −0.70 | −0.82 | −1.20 | −0.78 | −0.76 | −1.20 | −1.18 | −0.81 | −0.90 | −0.90 | −0.94 |
| | Deviation % | 3.96 | −2.78 | 1.23 | 6.19 | −7.14 | 0.00 | 1.69 | 5.36 | −5.81 | −2.17 | −2.17 | 5.62 |
| The second-stage adjustment (26 + 34 + 42) | Theoretical value | 3.82 | 4.11 | 4.03 | 3.71 | 4.01 | 3.99 | 3.89 | 3.92 | 3.92 | 3.92 | 3.92 | 3.94 |
| | Measured value | 4.02 | 3.93 | 4.08 | 3.90 | 3.84 | 4.03 | 4.03 | 4.12 | 3.71 | 3.79 | 3.96 | 4.13 |
| | Deviation % | 5.24 | −4.38 | 1.24 | 5.12 | −4.24 | 1.00 | 3.60 | 5.10 | −5.36 | −3.32 | 1.02 | 4.82 |
| The third-stage adjustment (50 + 58 + 66) | Theoretical value | 3.88 | 4.17 | 4.03 | 3.94 | 3.98 | 3.98 | 3.97 | 3.97 | 3.97 | 3.97 | 3.98 | 3.99 |
| | Measured value | 4.10 | 4.00 | 4.11 | 4.12 | 3.75 | 4.03 | 4.14 | 4.20 | 3.70 | 3.85 | 3.94 | 4.20 |
| | Deviation % | 5.67 | −4.08 | 1.99 | 4.57 | −5.78 | 1.26 | 4.28 | 5.79 | −6.80 | −3.02 | −1.01 | 5.26 |
| The fourth-stage adjustment (74 + 82 + 90) | Theoretical value | 1.75 | 1.75 | 1.75 | 1.75 | 1.75 | 1.75 | 1.75 | 1.75 | 1.75 | 1.75 | 1.75 | 1.75 |
| | Measured value | 1.83 | 1.68 | 1.78 | 1.82 | 1.66 | 1.77 | 1.81 | 1.79 | 1.69 | 1.70 | 1.77 | 1.82 |
| | Deviation % | 4.57 | −4.00 | 1.71 | 4.00 | −5.14 | 1.14 | 3.43 | 2.29 | −3.43 | −2.86 | 1.14 | 4.00 |

### 4.2.3. Evaluations of Different Pre-Stress Deviation Adjustment Schemes

To compare the effects of different pre-stress deviation adjustment schemes, the results of pre-stress deviations and shape deviations produced by the following five adjustment schemes were further analyzed and compared in this paper. The five schemes were as follows (as shown in Figure 5):

1.　Scheme 1: 12 DC1 were used to adjust the deviations of the 12 measuring points, as mentioned above.
2.　Scheme 2: Only 10 DC1 (No. 2, 10, 18, 26, 42, 50, 58, 66, 82, and 90) were used to adjust the deviations of the 12 measuring points.
3.　Scheme 3: Only 8 DC1 (No. 2, 10, 26, 34, 50, 58, 74, and 82) were used to adjust the deviations of the 12 measuring points.
4.　Scheme 4: Only 6 DC1 (No. 2, 18, 34, 50, 66, and 82) were used to adjust the deviations of the 12 measuring points.
5.　Scheme 5: Only 4 DC1 (No. 2, 26, 50, and 74) were used to adjust the deviations of the 12 measuring points.

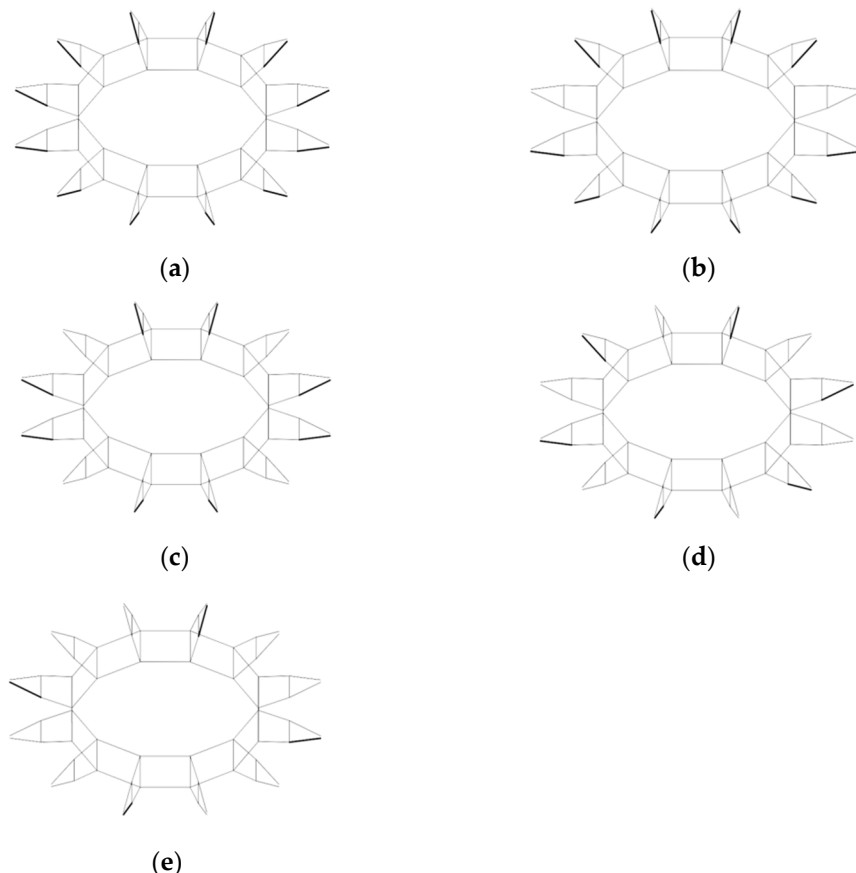

(a)　　　　　　　　　　　　　　　(b)

(c)　　　　　　　　　　　　　　　(d)

(e)

**Figure 5.** Five different schemes with different adjustment cables; (**a**) scheme 1; (**b**) scheme 2; (**c**) scheme 3; (**d**) scheme 4; (**e**) scheme 5.

The adjustment results were presented in Table 4, as follows:

1.　When 12 DC1 were used for adjustment, the number of adjustment cables equaled to the number of measuring points. Then Equation (11) can be used to obtain the unique length adjustment, $\delta e_0^k$ (as shown in Table 2), and the only accurate correction value of internal force, $\delta \hat{t}^k$, can be further obtained (as shown in Table 4) with $\eta = 0$. As the number of adjustment cables decreased, that is, the number of adjustment cables was

lower than that of the measuring points, the approximate solution, $(\delta e_0^k)^{\#}$, can only be obtained by using the least square method and Equation (12).

2. The effects of pre-stress deviation adjustment decreased with the decrease of the number of adjustment cables. The adjustment effects evaluation index of internal force, $\eta$, gradually increased from 0 to 0.4439 when the number of adjustment cables were reduced from twelve to four.

3. As the number of adjustment cables decreased, the overall structural deformation caused by the pre-stress deviation adjustment in this paper also gradually decreased. The deformation index, $\phi$, gradually decreased from $94.049 \times 10^{-4}$ to $1.384 \times 10^{-4}$ when the number of adjustment cables decreased from twelve to four. It indicated that the pre-stress deviations of the twelve measuring points were adjusted to zero through twelve adjustment cables, yet it generated larger deformation in comparison with the schemes that had fewer adjustment cables.

4. In addition, this study revealed that the same number of adjustment cables with different adjustment sequences would lead to different pre-stress deviations and shape deformations; however, the adjustment results remained the same. In this case, it was necessary to further optimize the adjustment scheme that produced lower peak values of internal force and shape change during the adjustment process.

**Table 4.** Adjustment results of five adjustment schemes (kN).

| Measuring Point | $\delta t_q$ | Scheme 1 | | Scheme 2 | | Scheme 3 | | Scheme 4 | | Scheme 5 | |
|---|---|---|---|---|---|---|---|---|---|---|---|
| | | $\hat{\delta t}^k$ | $\delta$ | $\hat{\delta t}^k$ | $\delta * 10^{-3}$ | $\hat{\delta t}^k$ | $\delta$ | $\hat{\delta t}^k$ | $\delta$ | $\hat{\delta t}^k$ | $\delta$ |
| 2 | −0.120 | 0.120 | 0.000 | 0.111 | 0.089 | 0.081 | 0.002 | 0.244 | 0.015 | −0.062 | 0.033 |
| 10 | 0.245 | −0.245 | 0.000 | −0.228 | 0.294 | −0.305 | 0.004 | −0.101 | 0.021 | −0.025 | 0.048 |
| 18 | 0.151 | −0.151 | 0.000 | −0.061 | 8.113 | −0.001 | 0.023 | −0.126 | 0.001 | −0.029 | 0.015 |
| 26 | −0.178 | 0.178 | 0.000 | 0.195 | 0.294 | 0.113 | 0.004 | 0.023 | 0.024 | −0.056 | 0.055 |
| 34 | 0.110 | −0.110 | 0.000 | −0.119 | 0.089 | −0.179 | 0.005 | −0.091 | 0.000 | −0.029 | 0.007 |
| 42 | 0.189 | −0.189 | 0.000 | −0.205 | 0.250 | −0.063 | 0.016 | −0.109 | 0.006 | −0.025 | 0.027 |
| 50 | −0.234 | 0.234 | 0.000 | 0.219 | 0.227 | 0.159 | 0.006 | 0.237 | 0.000 | −0.061 | 0.087 |
| 58 | −0.165 | 0.165 | 0.000 | 0.151 | 0.189 | 0.069 | 0.009 | −0.072 | 0.056 | −0.022 | 0.035 |
| 66 | 0.154 | −0.154 | 0.000 | −0.169 | 0.236 | −0.109 | 0.002 | −0.225 | 0.005 | −0.056 | 0.010 |
| 74 | −0.218 | 0.218 | 0.000 | 0.204 | 0.189 | 0.128 | 0.008 | 0.100 | 0.014 | 0.040 | 0.032 |
| 82 | 0.312 | −0.312 | 0.000 | −0.327 | 0.227 | −0.356 | 0.002 | −0.267 | 0.002 | −0.056 | 0.065 |
| 90 | 0.198 | −0.198 | 0.000 | −0.214 | 0.250 | 0.031 | 0.053 | −0.062 | 0.019 | −0.022 | 0.031 |
| $\eta$ | | 0 | | 0.0104 | | 0.1320 | | 0.1630 | | 0.4439 | |
| $\phi * 10^{-4}$ | | 94.049 | | 87.712 | | 51.690 | | 46.500 | | 1.384 | |

## 5. Conclusions

Considering the fact that there was a lack of an efficient method to analyze and evaluate the construction errors of the cable-bar tensile structure, and especially as no proper adjustment method was available to correct these errors, construction errors analysis, construction errors adjustment method, and evaluation of the adjustment efficiency were studied in this paper. The research results demonstrated that: (1) The adjusted pre-stress deviations of measuring points can be effectively corrected, and the calculated results were coincided with the experimental results, indicating the feasibility and validity of proposed error analysis and adjustment method of pre-stress deviation of actual pretension structures. (2) The adjustment effects of pre-stress deviations varied from the number of adjustment members. When the number of adjustable members equaled to the number of measuring points, accurate adjustment of pre-stress deviation can be achieved. When the number of adjustment members was lower than the number of measuring points, it was impossible to accurately adjust the pre-stress deviation of measuring points. Additionally, the adjustment effectiveness of adjustment gradually decreased with the decrease in the number of adjustment members. (3) During the process of pre-stress deviation adjustment, different adjustment schemes produced different structural deformations. In this study, as the number of adjustment cables decreased from twelve to four gradually, the overall struc-

tural deformation caused by the pre-stress deviation adjustment also gradually decreased. (4) The same number of adjustment cables with different adjustment sequences would lead to different pre-stress deviations and shape deformations. Thus, it was necessary to further optimize the adjustment method that resulted in lower peak values of internal forces and shape changes during the adjustment process.

**Author Contributions:** Conceptualization, L.C., methodology, L.C.; software, Y.L.; validation, Y.Z. (Yihong Zeng) and H.Z.; resources, L.C. and Y.Z. (Yiyi Zhou); writing—original draft preparation, L.C.; writing—review and editing, L.C.; supervision, L.C.; project administration, L.C.; All authors have read and agreed to the published version of the manuscript.

**Funding:** This research was funded by National Natural Science Foundation of China (Grant No. 51578422, No. 51678082).

**Institutional Review Board Statement:** Not applicable.

**Informed Consent Statement:** Not applicable.

**Data Availability Statement:** Not applicable.

**Conflicts of Interest:** The authors declare no conflict of interest.

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
