# Peer review of "Theoretical Analysis and Experimental Research on the Adjustment for Pre-Stress Deviation of the Cable-Bar Tensile Structures"

_applsci, doi:10.3390/app11125744_

Round 1
Reviewer 1 Report
This reviewed manuscript is a well elaborated scientific-research work oriented to adjustment for pre-stress deviation of the cable-bar tensile structures. The manuscript fulfils the criteria required for publishing in the given professional journal. I appreciate an overview of the issues presented in the chapter Introduction as well as the theoretical background of the applied fundamental theory, which is introduced in the chapters No.2 and No.3. The focus of the whole scientific investigation consists in the experimental measurements that were performed on the really designed and constructed cable-bar tensile structure model. I do not have any special comments concerning improvement of this manuscript, but I have the following recommendations:
- it could be useful for the readers, for a better understanding, to explain a meaning of the abbreviations HC, RC, DC as well as to describe a difference between the active members and passive members in the model, chapter 4.1
I suggest to accept this manuscript after taking into account my recommendations.
Reviewer 2 Report
The manuscript presents an interesting theoretical method with experimental validation on control of applied forces on cabled structures. Such study is utmost importance and could potentially provide further help to practicians with regards to assembling complex tensile roof structures. The findings have been encouraging, albeit it could perhaps be a bit more explored. A throughout proofreading is also recommended. The following comments are suggested for consideration prior to the publication of the article:
- Please, provide sketches with the identification of the cables adjusted in each scheme. For example, for Scheme 5 on Table 4, a sketch highlighting where cables 2, 26, 50 and 74 are, would be extremely helpful for the reader.
- Is there any other set of adjusted cables (for example, another set of four cables) that could produce a higher “adjustment effects evaluation index of internal force” than 0.4439?
- Any advice for further studies on how to find the optimum adjustment sequence?
- Throughout proofreading:
- To avoid repetition, such as in “… pre-stress distribution of the structure to optimize stiffness distribution of the structure, …”
- To avoid vagueness, such as in the abstract “… an adjustment method was devised for the pre-stress deviations under various conditions and…”. Which conditions? Conditions related to what? Why were these conditions chosen?
- To avoid misinterpretation, such as in “node size error, etc”. What you mean by a size error in the node? Wouldn’t be related to “position” rather than “size”? Please clarify.
Reviewer 3 Report
The paper should be improved:
1 in the introduction it should be explained clearly what it is the new the paper related to the previous work of the same authors
2 Some formulas in pages 3, 4 (Section 2) should be explained better. In particular there is no reference to symbols n, b…etc. In addition the formulas should be written according to the rules of the Journal.
4 The imposed boundary conditions should be explained better and in the model a reference should be made for the number of the adjustment members used in order to have convergence for the results.
5 In the experimental procedure the nomenclature introduced HC1… RC1, DC1...should be explained.
6 The conclusions Section should be limited. On the other hand it should be mentioned what are the benefits of the proposed analysis related to previous studies.
Round 2
Reviewer 3 Report
The most of my remarks have been made.